# Body Image among Girls in Indonesia: Associations with Disordered Eating Behaviors, Life Engagement, Desire for Cosmetic Surgery and Psychosocial Influences

**DOI:** 10.3390/ijerph20146394

**Published:** 2023-07-19

**Authors:** Kirsty May Garbett, Nadia Craddock, L. Ayu Saraswati, Phillippa C. Diedrichs

**Affiliations:** 1Centre for Appearance Research, University of the West of England, Coldharbour Lane, Bristol BS16 1QY, UK; nadia.craddock@uwe.ac.uk (N.C.); phillippa.diedrichs@uwe.ac.uk (P.C.D.); 2Department of Women, Gender, and Sexuality Studies, University of Hawai’i at Mānoa, Honolulu, HI 96822, USA

**Keywords:** body image, mental health, self-esteem, disordered eating, Indonesia, Asia

## Abstract

Body image is an important risk factor for mental and physical health during adolescence. Nonetheless, few studies have empirically investigated body image among girls in Southeast Asia. This cross-sectional study examined the prevalence of body esteem (a holistic measure of body image assessing the degree of satisfaction with one’s appearance, weight, and shape), associated behavioral outcomes, and potential psychosocial influences on body image among Indonesian girls to inform future intervention efforts. Girls across Indonesia with internet access (N = 318, M_age_ = 13.59 years) completed a questionnaire that assessed body image, disordered eating behaviors, appearance-related life engagement, desire for cosmetic surgery, appearance ideal internalization, media literacy, appearance-related teasing, and self-esteem. The younger girls’ (10–13 years) responses were analyzed separately from those of the older girls (14–17 years). Over half of the girls did not have high body esteem. Approximately a quarter of the girls across both age groups reported restricting their food intake, emotionally eating, and/or engaging in excessive exercise, and over half desired cosmetic surgery. Disordered eating behaviors and a desire for cosmetic surgery were negatively correlated with body esteem. Hierarchical regression analyses found BMI, internalization, appearance-related teasing, and self-esteem were uniquely associated with body esteem among younger girls. Appearance-related teasing and self-esteem were positively associated with body esteem among older girls. These findings suggest body image interventions may be important for this population, with appearance-related teasing and self-esteem potentially useful targets for intervention efforts. Further prospective research to investigate these findings would be beneficial for a deeper understanding of body image risk factors for girls across Indonesia.

## 1. Introduction

Body image refers to how a person thinks, feels, and behaves toward their body [1]. The pre-adolescence and adolescence stage is a critical period for body image development due to the substantial physical, psychological, and social changes experienced at this time [2,3,4]. Notably, research indicates that girls fare worse than boys in the severity with which they experience negative body image [4,5]. Although research indicates that negative body image is highly prevalent across high-, middle-, and low-income countries [6], research efforts are overwhelmingly concentrated across high-income, English-speaking countries [7], mirroring mental health research more broadly [8]. Scholars across the field of mental health strongly advise against reliance on research conducted in high-income countries in the identification and treatment of mental health conditions in other parts of the world [8]. Therefore, the current research sought to examine the associations of some key behavioral and psychosocial constructs with body image among girls across multiple provinces of Indonesia in order to inform preventative efforts in this setting.

### 1.1. Body Image among Indonesian Girls

Indonesia is the fourth most populous country in the world, yet the extant literature exploring body image among girls in this context is disproportionately sparse. Current research indicates Indonesian girls and young women report feeling appearance pressure from the media in relation to their weight [9] and their skin shade [10]. Specifically, girls feel pressure to be thin and report disordered eating attitudes and behaviors [11]. Likewise, girls often desire a lighter skin shade, which is corroborated by the prolific use of skin-lightening products across Indonesia [12]. Quantitative research suggests that Indonesian girls’ feelings of self-worth are negatively correlated with body image concerns [13]. Moreover, limited research suggests young women in Indonesia avoid social events due to appearance-related concerns [14] and, according to a study across five Southeast Asian countries, including Indonesia [15], just over a quarter of young women desire cosmetic surgery to change aspects of their appearance they do not like.

The research presented thus far is concentrated in just a few of Indonesia’s 34 provinces, exclusively on the island of Java. Given the rich cultural and ethnic diversity across Indonesia, it is important that this is reflected in research. Furthermore, research to date exclusively focuses on older adolescent girls and young women. In other contexts, including other parts of Asia [16], research shows that girls can begin to experience body image concerns from early adolescence. Thus, an understanding of when body image concerns emerge among Indonesian girls is an important consideration in order to identify the optimal time for preventative action.

The behavioral epidemiological framework offers a useful lens from which to view the sequential research steps necessitated in the development of health intervention and prevention efforts [17]. Within this framework, two of the earliest phases of research development involve (a) establishing a link between a construct (i.e., body image) and behavior and (b) identification of factors that influence a construct. It is only after these are established that a sound rationale can be made for proceeding toward health promotion and prevention efforts. As previously stated, it is not enough to assume that the relationships between constructs in one region or cultural context will replicate in the same manner in a new cultural context [8].

### 1.2. The Present Study

The current research sought to bolster our understanding of body image among Indonesian girls, specifically targeting the two early phases of the behavioral epidemiological framework in a stepwise approach toward the development of culturally sensitive intervention efforts. The first research aim was to explore the prevalence of body image concerns and related constructs among Indonesian girls and examine the relationship between these constructs. Specifically, theoretically and empirically supported adverse outcomes of negative body image in other geographical contexts were investigated: namely, disordered eating behaviors, appearance-related life engagement, and desire for cosmetic surgery [18,19,20]. A richer understanding of body image concerns and its relationship with a number of key behavioral and psychosocial constructs among girls in Indonesia is important as it may prove influential in advancing preventative efforts at a community (e.g., schools) and/or government level [21]. It was hypothesized that body esteem—a holistic measurement of body image that assesses the degree of satisfaction with one’s appearance, weight, and shape—would be negatively associated with disordered eating behaviors, desire for cosmetic surgery, and disengagement with important life activities.

The second research aim was to consider the relationship between body image and established psychosocial influences of body image in other cultural contexts, namely, internalization of appearance ideals, media literacy, appearance-related teasing, and self-esteem. These influences are underpinned by two theoretical frameworks. Firstly, the Tripartite Influence Model [1], which stresses the importance of sociocultural influences such as the media, peers, and family members (all of which can be a source of appearance-related teasing), as well as the extent a person internalizes the appearance ideals of society. Secondly, the Biopsychosocial Model [22] stresses the importance of broader psychological factors such as self-esteem. Each of these factors has been identified as having important influences on body image among adolescents [23,24], with greater internalization of appearance ideals, lower media literacy skills, higher perceived appearance-related teasing, and lower self-esteem related to greater body image concerns. Moreover, these factors have been identified as modifiable targets in the prevention of body image concerns in this age group [25]. A deeper understanding of the relevance of these factors among Indonesian girls is important to inform the foundations of a framework from which to build preventative efforts. It was hypothesized that internalization of appearance ideals, media literacy, appearance-related teasing, and self-esteem would be significantly associated with body image among Indonesian girls. Specifically, lower internalization and appearance-related teasing, and higher media literacy and self-esteem would be associated with higher body esteem.

Further, we examined these research aims separately among both younger (10–13 years) and older girls (14–17 years). Research to date in Indonesia has exclusively focused on adolescent girls. However, research from other contexts suggests body image concerns present earlier than this [4]. Understanding if such concerns, and their relationship with key health behaviors and psychosocial influences, are present among a younger cohort would be useful to guide if and when to intervene with preventative efforts in the Indonesian context.

## 2. Materials & Methods

### 2.1. Participants and Procedure

The Dove Global Girls Beauty and Confidence Study is a global study commissioned by the Dove Self-Esteem Project. The purpose of the study was to conduct a detailed exploration of body image attitudes of girls around the world aged 10–17 years using nationally representative samples of girls with internet access. A total of 5165 girls across 18 countries took part. This paper reports only on the responses collected in January 2017 from 318 Indonesian girls from across 24 of Indonesia’s largest provinces (there is a total of 34 provinces across Indonesia).

Participants were recruited via a research agency panel in which participants’ parents were previously registered. Parents received an email invitation or responded to an advert for the study on the research agency’s website. After showing initial interest, informed parental consent (for participants under 16 years old) and participant assent was obtained. Participants completed an online questionnaire at home and received research agency points for participation, which they could redeem for goods or shopping vouchers.

### 2.2. Measures

The same measures were used for younger (10–13 years) and older (14–17 years) girls unless otherwise stated.

Participant characteristics. On behalf of participants, parents reported annual household income and the region of Indonesia where they live. Age and self-reported weight and height were collected from participants. BMI was calculated as weight/height^2^.

Body image. The weight and appearance subscales of the Body Esteem Scale for Children (BES-C) were used to measure body image among younger girls [26]. The scale consisted of 16 items (for example, the item, ‘*I like what I see when I look in the mirror*’). To aid comprehension, the scale was adapted to measure responses on a four-point Likert scale (1 = yes, a lot − 4 = no, not at all). A continuous score was derived, with items reverse scored so that higher values indicated higher body esteem. Due to a high correlation between the subscales (*r* = 0.76), the subscales were collapsed to create a total body esteem score. Reliability for the scale was good (α = 0.79).

The weight and appearance subscales of the Body Esteem Scale for Adults and Adolescents (BESAA) were used among older girls [27]. This scale consisted of 18 items (for example, the item, ‘*I worry about the way I look*’). The scale is scored on a five-point Likert scale (1 = never − 5 = always), with some items reverse scored so that higher values indicated higher body esteem. As with the younger girls, due to a high correlation between the subscales (*r* = 0.73), a total body esteem score was created. Reliability for the scale was good (α = 0.86).

To create prevalence scores of body esteem, possible scores were divided equally into three categories: high, medium, or low body esteem. For younger girls, mean scores on the BES-C between 1–2 were categorized as “low”, 2–3 as “medium”, and 3–4 as “high” body esteem. For older girls, mean scores on the BESAA between 1–2.33 were categorized as “low”, 2.34–3.66 as “medium”, and 3.67–5 as “high” body esteem.

Disordered eating behaviors. Four binary items were used to measure disordered eating behaviors. Participants were asked if they had engaged in the following because they didn’t feel good about the way they looked: restrictive eating (‘*changed what I ate to cut out certain food groups or skip meals*’), food avoidance (‘*stopped myself from eating even when I felt hungry*’), excessive exercise (‘*exercised more than is healthy in an effort to lose weight*’), or emotional eating (‘*binged or eaten food to make myself feel better*’ or ‘*binged or eaten food for comfort*’ for younger and older girls, respectively). Analyses were conducted on these behaviors individually as well as on total disordered eating behavior scores, computed as the sum of the four binary variables, which ranged from 0 (no disordered eating behaviors) to 4 (four disordered eating behaviors).

Life engagement. Life engagement was measured using an adapted version of the Body Image Life Disengagement Questionnaire [21]. This scale asks, ‘*Which of the following have you NOT done because you didn’t feel good about the way you looked?*’ followed by ten activities (e.g., ‘*gone to a social event, party, or club*’, ‘*given an opinion*’). Participants selected all activities they had chosen to disengage with due to appearance-related concerns (‘yes/no’). A total score was created to reflect life disengagement, with higher scores reflecting more life disengagement.

Desire for cosmetic surgery. The girls’ desire for cosmetic surgery was measured using a single item, ‘*If I had the time and/or money I would have cosmetic surgery to correct aspects of my body I do not like’*, using a five-point Likert scale (1 = never − 5 = always). Higher scores indicated more desire for cosmetic surgery. Only older girls were provided with this item as it was deemed inappropriate to ask this question of 10–13-year-olds. To create a prevalence score, participants who reported considering cosmetic surgery at least ‘rarely’ (a score of 2+) were categorized as having ‘cosmetic surgery desire’.

Internalization. Internalization of appearance ideals was measured using a single item (‘*I try my best to look like the models and/or celebrities I see in the media*’) using a five-point Likert scale (1 = strongly disagree − 5 = strongly agree). Higher scores indicated higher internalization.

Media literacy. Media literacy was measured using three items on a five-point Likert scale (1 = strongly disagree − 5 = strongly agree). The items were: ‘*I know that images of girls and women in the media are digitally altered or airbrushed*’; ‘*Very few real women and girls look like the women and girls in adverts, movies, television etc.*’; and ‘*Most people can’t look like the women and girls shown in the media*’. Scores across these items were averaged, with higher scores reflecting greater critical thinking about media messages. Reliability was acceptable for a three-item scale for both younger (α = 0.61) and older (α = 0.64) girls.

Appearance-related teasing. Appearance-related teasing was measured using a single item, ‘*Other people make fun of the way I look*’ (1 = no, not at all − 4 = yes, a lot) and, ‘*I get teased or bullied because of the way I look*’ (1 = never − 5 = always) for younger and older girls, respectively. For both age groups, higher scores indicated more appearance-related teasing.

Self-esteem. Self-esteem was measured using the Rosenberg Self-Esteem Scale [28]. Participants responded on a four-point Likert scale (1 = strongly disagree − 4 = strongly agree), with some items scored in reverse so that higher scores indicated higher self-esteem. Internal reliability for the ten-item scale was acceptable for older girls (α = 0.70) and slightly lower than acceptable for younger girls (α = 0.64). Reliability analyses indicated that the removal of item 8 (‘*I wish I could have more respect for myself*’) improved the reliability of the scale across both groups (α = 0.74 and α =0.79 for younger and older girls, respectively). Therefore, this item was removed from the scale for both groups.

### 2.3. Analyses

All analyses were conducted using SPSS, Version 26. The normality of continuous variables was assessed using histograms, skewness, and kurtosis. All variables except appearance-related teasing met normality assumptions. Square root transformations brought the teasing measure in line with normality assumptions. Four univariate outliers were identified. Data were analyzed using the untransformed and transformed variables, as well as with outliers included and removed. The significance of the results did not vary, so untransformed data with outliers included are presented.

Missing data ranged from 0–0.6%; therefore, listwise deletion was deemed appropriate. Mean and standard deviations for continuous variables and percentages for categorical variables are presented.

Bivariate correlations examined the association between body esteem and disordered eating behaviors, life engagement, cosmetic surgery desire, and hypothesized psychosocial influences. Following this, hierarchical linear regressions examined the unique impact of each hypothesized psychosocial influence on body esteem after controlling for age and BMI. The data did not violate assumptions of normality, linearity, multi-collinearity, or homoscedasticity. All analyses were conducted separately for younger and older girls.

## 3. Results

### 3.1. Sample Characteristics

The mean age was 11.58 years (*SD* = 1.10) for the younger girls and 15.51 years (*SD* = 0.97) for the older girls. Annual household income was measured as an assessment of socioeconomic status, which ranged from under 10,000,000 to over 200,000,000 Indonesian rupiah. The mean annual household income was between 70,000,000–79,999,999 Indonesian rupiah, which is representative of the average household income in Indonesia in 2017 [29]. The average self-reported BMI of girls was 21.2 (*SD* = 5.59).

### 3.2. Prevalence of Body Esteem, Disordered Eating Behaviors, Life Engagement and Cosmetic Surgery Desire

Table 1 presents the prevalence of body esteem, disordered eating behaviors, life engagement and cosmetic surgery desire among younger and older girls. Almost half the girls in both age groups were categorized as having high body esteem, with a similar number expressing medium body esteem. Very few (*n* = 2) were categorized as having low body esteem. Over a fifth of girls in both the younger and older age groups reported engaging in each disordered eating behavior, with the exception of food avoidance among older girls, where only 11.7% reported doing so. Between 4.5–21.9% of younger girls and 4.9–30.1% of older girls opted out of each life event due to appearance-related concerns. The percentage of girls opting out of at least one activity due to appearance-related concerns was 51.6% in the younger group and 63.2% in the older group. Lastly, 54.6% of older girls reported a desire for cosmetic surgery.

### 3.3. Relationship between Body Esteem and Disordered Eating Behaviors, Life Engagement, and Cosmetic Surgery Desire

Table 2 presents the correlations between body esteem and disordered eating behaviors, life engagement and desire for cosmetic surgery. Small negative correlations were found between body esteem with food restriction and food avoidance for both age groups, indicating that those with higher body esteem were less likely to restrict or avoid food. Small-to-moderate negative correlations were also found between body esteem and excessive exercise and emotional eating for younger girls only. This indicates that those with higher body esteem were less likely to exercise excessively in an attempt to lose weight or eat emotionally. Moderately negative correlations between body esteem and total disordered eating behavior scores were found for younger girls only. A small-to-moderate negative correlation was found with respect to body esteem and life engagement in both age groups, suggesting those with higher body esteem are less likely to be held back by appearance-related concerns. Lastly, analyses of the older girls showed a moderate negative correlation between body esteem and desire for cosmetic surgery, such that higher body esteem was related to less desire for cosmetic surgery. All correlations were in the anticipated direction.

### 3.4. Relationship between Psychosocial Influences and Body Esteem

Table 2 presents the correlations between psychosocial influences and body esteem. Body esteem showed a small-to-moderate negative correlation with internalization and a moderate negative correlation with appearance-related teasing for both age groups. Body esteem showed a moderate and strong positive correlation with self-esteem for younger and older girls, respectively. These correlations were all significant at the *p* < 0.01 level. There was no correlation between body esteem and media literacy for either age group.

The hierarchical linear regression analyses are reported in Table 3. For younger girls, after controlling for age and BMI (step 1), the addition of psychosocial influences (step 2) accounted for an additional 52% of the variance in body esteem. BMI, appearance-related teasing, internalization, and self-esteem emerged as unique significant predictors, indicating that having a lower BMI, lower internalization of appearance ideals, having higher self-esteem, and experiencing less appearance-related teasing were associated with higher body esteem. For older girls, after controlling for age and BMI (step 1), the addition of psychosocial influences (step 2) accounted for an additional 65% of the variance in body esteem. Appearance-related teasing and self-esteem emerged as unique significant predictors, indicating that experiencing less appearance-related teasing and having higher self-esteem were associated with higher body esteem in this age group.

## 4. Discussion

The aim of this study was to explore body image among Indonesian girls, specifically examining the prevalence of body esteem (a holistic measure of body image, assessing the degree of satisfaction with one’s appearance, weight, and shape) and its association with disordered eating behaviors, life engagement, desire for cosmetic surgery and psychosocial influences in this under-researched population. A nationally representative sample of girls in Indonesia who have access to the internet was drawn upon, providing an unprecedented snapshot of girls’ body image in Indonesia.

### 4.1. Prevalence of Body Esteem and Related Constructs

Approximately half of the girls experienced high body esteem; the remaining half experienced medium body esteem. Just two girls in the study experienced low body esteem. This finding is in line with research in other contexts. For example, in the UK, around two-thirds of early adolescent girls express medium-to-low body esteem, with one-third expressing high body esteem [25]. Similar rates of body image concerns have been found among girls in India [30], Thailand [31], and Brazil [32]. While measurement differences make cross-cultural comparisons difficult, collectively, these studies suggest the prevalence of body esteem among Indonesian girls is akin to elsewhere.

This study sought to understand the prevalence of behavioral outcomes previously found to be associated with body image in other populations [15,20,33]. Four disordered eating behaviors were found to be prevalent, with emotional eating being the most prevalent (31.1%), followed by excessive exercise (26.4%), food restriction (23.9%), and food avoidance (16.7%). This level of prevalence is similar to rates among adolescent girls elsewhere, such as in the US [34], China [35], and Malaysia [36]. This study adds to the growing evidence that disordered eating behaviors are comparably prevalent in Asian countries as they are in Europe and North America [37].

The extent girls may disengage from life events due to appearance-related concerns was also considered, as has been reported in previous qualitative work with Indonesian women [10]. Over half of girls reported disengaging with at least one life activity due to appearance concerns. This finding underscores the importance of interventions to alleviate body image concerns among girls in Indonesia, given the wide-ranging impact appearance concerns can have in this demographic in terms of education (i.e., 14–15% of girls failing to raise their hands in class), social relationships (i.e., 30% of older girls opting out of social events), and health (i.e., 12–13% neglecting visits to a doctor or nurse).

The desire for cosmetic surgery among Indonesian adolescent girls was investigated, a relatively unexplored area in this population. High rates of the desire for cosmetic surgery among young Indonesian women [15] and the rise of the industry across urban sites in Indonesia [38] suggest that adolescents may also share a desire for cosmetic surgery. Indeed, this was found to be the case: over half of girls aged 14–17 years agreed that if time and money were not a concern, they would have cosmetic surgery. This finding indicates that adolescents are unhappy with aspects of their appearance and willing to undertake procedures that are associated with great expense and risk of medical complications to change them. This is somewhat contradictory to qualitative research, which found that for religious and cultural reasons, as well as a fear of pain, many Indonesian women do not desire cosmetic surgery [39]. This study suggests that Indonesian girls may be less influenced by religious and cultural factors when considering cosmetic surgery than older women or perhaps have not considered the risks involved. This area warrants further research, particularly as invasive surgeries, such as rhinoplasty and blepharoplasty, are conducted on Indonesian adolescents only slightly older than those reported in this study [38].

### 4.2. Relationship between Body Image and Disordered Eating, Life Engagement and Cosmetic Surgery Desire

Low body esteem was related to greater food restriction and food avoidance across both age groups studied. In addition, low body esteem was associated with excessive exercise with the goal of losing weight and emotional eating for the younger age group. Among older girls, low body esteem was associated with greater disengagement from life activities due to appearance-related concerns and a greater desire for cosmetic surgery. These findings are in line with research conducted among young university-aged Indonesian women [11,14,40] and align with research in other countries and cultures [20,33]. As such, these findings suggest body image may be an important variable to consider when seeking to reduce patterns of disordered eating, increase educational engagement, tackle health inequalities, and address appearance-altering surgery uptake among Indonesian girls (although causality cannot be determined from the present findings alone). Together, these bolster previous findings of this nature in Indonesia and add to the earliest phase of the behavioral, epidemiological framework (i.e., establishing the impact of a construct) in relation to girls in Indonesia.

Contrary to the hypotheses, this study did not find an association between body esteem and excessive exercise and emotional eating among older adolescent girls, and accordingly, a non-significant relationship was found between body esteem and total disordered eating score in this age group. While completing the BESAA, it may be that older girls were focused on other aspects of their appearance salient in Indonesian culture (e.g., skin shade; [10]) rather than weight, hence the reason for the lack of relationship with excessive exercise and emotional eating. Relatedly, BMI was not found to be significantly associated with body esteem among older girls. Again, this is suggestive that other facets of appearance beyond weight may be influential in older adolescents’ body esteem. Alternatively, it may be that weight is not a concern among older adolescent girls in Indonesia, although several studies find this not to be the case [9,41]. A replication study that utilizes a body image measure that focuses solely on weight and shape, such as the shape and weight concerns subscales of the Eating Disorder Examination Questionnaire [42], may further our understanding of this finding.

### 4.3. Body Image and Psychosocial Influences

The second research aim sought to investigate how psychosocial factors relate to body image concerns among Indonesian girls in order to identify targets for future intervention efforts. Hierarchical regression analyses found that the modifiable influences under investigation (namely, internalization of appearance ideals, appearance-related teasing, and self-esteem) accounted for a large proportion of the variance in body esteem. Specifically, these factors explained 52% of the variance in body esteem in the younger age group and 65% of the variance in the older age. This is a noteworthy finding as it suggests intervention efforts targeting these factors may be successful among this population. Appearance-related teasing and self-esteem emerged as unique independent predictors of body esteem among younger and older girls. This finding, along with the high prevalence of teasing among adolescents in Indonesia [43], indicates that this factor may be a worthwhile target for future intervention efforts, along with self-esteem. Interventions targeting these factors elsewhere show promise in improving body image [44,45,46] and may provide the foundations for fruitful avenues of intervention research in Indonesia.

Contrary to hypotheses, media literacy was not associated with body esteem for either age group, nor did it account for any variance in body esteem during hierarchical regression analyses. This may have been due to the lack of variability in media literacy levels in the sample, as evidenced by the low standard deviations, or it may have been an inappropriate measure. A lack of valid measurement of appearance-related media literacy is an ongoing setback in the field of body image generally, and the need for measurement tools to assess specific aspects of media literacy has been well documented [47]. The lack of relationship between media literacy and body esteem may highlight a relatively low influence the media has on girls’ body image in this context. This study identified teasing (from friends or family members, face-to-face or via social media) as much more strongly correlated with body esteem than internalization, another key indicator of media pressure.

### 4.4. Limitations, Strengths and Future Directions

This study must be viewed in light of a number of limitations. First, although the internal consistency was deemed acceptable across all multi-item measures, a lack of valid measures among Indonesian youth dictated an over-reliance on single-item measures and measures validated among other populations (e.g., the BESAA is validated among Canadian adolescent girls, [27]). Measures of body image and related constructs are urgently required to strengthen understanding of these topics in this population. Second, as this study was cross-sectional in nature, causal relationships can only be hypothesized. With the exception of self-esteem, which likely has a reciprocal, bi-directional relationship with body image [23], the direction of the relationships hypothesized in this study is well-established both in terms of theory (e.g., Tripartite Influence Model, [1]) and in experimental and prospective research in other cultural contexts [19,48,49]. As such, we can be cautiously confident in our interpretations of the findings presented here. Nevertheless, determining causal relationships using prospective research designs is required in the Indonesian context. Third, it should be noted that the data were collected in 2017 and thus might not be fully reflective of Indonesian girls’ body image and related factors today.

Regarding the strengths of this study, a representative sample of Indonesian girls who have internet access was utilized. In a country as geographically and culturally diverse as Indonesia [50], this is a significant strength. Future work conducted in Indonesia would benefit from obtaining representative samples, as well as unpicking the ways body image may manifest in different contexts within Indonesia. This study was also the first, to the authors’ knowledge, to quantitatively study the prevalence and correlates of body image concerns in pre-adolescent Indonesian girls, whom we know from other contexts are likely to experience significant difficulties regarding how they feel about the way they look [25].

This study has several implications. With respect to the research field, these findings further substantiate body image as a global issue among girls. Beyond utilizing longitudinal designs with validated measures for the population under study, future research would benefit from considering the unique impact and influences on body image both within and between contexts and cultures. For example, research has found that religion can be a protective factor against body image concerns [51]. As Indonesia is a highly religious and predominantly Muslim culture, this warrants further consideration. In terms of body image prevention work, this study is the first to suggest potential targets for intervention design for Indonesian girls, namely appearance-based teasing and self-esteem. This offers the opportunity to consider which existing evidence-based intervention programs targeting these factors may resonate in Indonesia with cultural adaptation. The study also suggests preventative efforts to tackle body image concerns in Indonesia may be useful among pre-adolescents, with the findings indicating that concerns are already established in this age group.

## 5. Conclusions

This study has identified that over half of Indonesian adolescent girls do not have high body esteem. It has highlighted a plethora of ways in which body esteem impacts adolescent girls in Indonesia, including problematic eating, disengagement with education, retreating from social situations, and desiring cosmetic surgery. This information is critical for advancing advocacy efforts, future funding, and implementation of strategies and interventions to promote body esteem among adolescents. This is particularly crucial in the context of Indonesia, where the provision of mental health services is sparse [52]. Studies such as this ensure that limited resources are channeled toward appropriate mechanisms for change. Continuing to understand the concerns and needs of girls in Indonesia with regard to their body image and potential intervention strategies is vital to ensure their potential is not stifled due to appearance-related concerns.

## Figures and Tables

**Table 1 ijerph-20-06394-t001:** Prevalence of body esteem, disordered eating behaviors, appearance-related life engagement and cosmetic surgery desire.

	Younger Age Group 10–13 Years Old		Older Age Group 14–17 Years Old	
	%	*n*	%	N
Body esteem				
High	47.7	74	50.3	82
Medium	51.6	80	49.1	80
Low	0.6	1	0.6	1
Disordered eating behaviors				
Food restriction	20.6	32	27	44
Food avoidance	21.9	34	11.7	19
Emotional eating	33.5	52	28.8	47
Excessive exercise	26.5	41	26.4	43
Life engagement				
Gone to the beach or pool, sauna, or spa	13.5	21	14.1	23
Gone to a social event or party	21.9	34	30.1	49
Gone shopping for clothes	10.3	16	8.6	14
Done a physical activity or sport	9.7	15	8	13
Given an opinion	6.5	10	11.7	19
Stood up for myself	9.7	15	11	18
Gone to school	4.5	7	4.9	8
Raised my hand up in class	14.8	23	14.1	23
Go to the doctor or school nurse	12.9	20	11.7	19
Socialized with my friends	9.7	15	12.3	20
Any of the above	51.6	80	63.2	103
Desire for cosmetic surgery	-	-	54.6	89

Note: Prevalence for life engagement items indicates the percentage and number of participants opting out of the activity due to appearance-related concerns.

**Table 2 ijerph-20-06394-t002:** Bivariate correlations between body esteem and disordered eating behaviors, cosmetic surgery desire and psychosocial factors.

	Younger Girls			Older Girls		
	M	SD	Body Esteem	M	SD	Body Esteem
Food restriction			−0.31 *			−0.16 *
Food avoidance			−0.28 *			−0.15 *
Excessive exercise			−0.39 *			0.03
Emotional eating			−0.21 *			0.00
Total disordered eating score	1.03	1.24	−0.42 *	0.94	1	−0.11
Life engagement	1.14	1.57	−0.3 *	1.26	1.46	−0.37 *
Cosmetic surgery desire	-	-	-	2.33	1.47	−0.59 *
Internalization	3.11	1.28	−0.35 *	3.06	1.29	−0.41 *
Media literacy	2.41	0.84	0.10	2.12	0.73	0.13
Teasing	3.10	0.93	−0.65 *	2.13	1.2	−0.67 *
Self-esteem	1.71	0.49	0.60 *	1.76	0.54	0.75 *

* Correlation is significant at the 0.01 level.

**Table 3 ijerph-20-06394-t003:** Hierarchical linear regressions for younger and older girls. The dependent variable is body esteem.

	Younger Girls					Older Girls				
	B	SE	*β*	t	*p*	B	SE	*β*	t	*p*
*Model 1*										
Age	0.02	0.03	0.04	0.52	0.61	−0.04	0.05	−0.06	0.75	0.45
BMI	−0.02	0.01	−0.33	−4.08	<0.001	0.02	0.01	0.12	0.12	0.14
R^2^	0.11					0.02				
Adjusted R^2^	0.09					0.00				
F (df)	8.46 (2) ^a^					1.31 (2)				
*Model 2*										
Age	0.02	0.02	0.05	0.87	0.39	0.02	0.03	0.04	0.76	0.45
BMI	0.01	0.04	0.18	3.32	<0.001	0.00	0.01	0.02	0.36	0.72
Internalization	−0.06	0.02	−0.18	−3.15	<0.01	−0.04	−0.02	0.10	−1.90	0.06
Teasing	−0.2	0.03	−0.41	−6.38	<0.001	−0.16	−0.03	−0.33	−5.63	<0.001
Self-esteem	0.33	0.06	0.37	6.09	<0.001	0.57	0.06	0.54	9.38	<0.001
R^2^	0.63					0.67				
Adjusted R^2^	0.61					0.65				
F (df)	46.22 (5) ^a^					59.65 (5) ^a^				

Note: ^a^
*p* < 0.001. B = unstandardized beta coefficients; SE = standard error; β *=* standard beta coefficient; R^2^ = R squared.

## Data Availability

The data that support the findings of this study are available from Edelman Intelligence, but restrictions apply to the availability of these data, which were used under license for the current study, and so are not publicly available. Data are, however, available from Edelman Intelligence upon reasonable request for non-commercial purposes. Contact Georgina Shan for more information-Georgina.Shann@edelmandataxintelligence.com.

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
