# Peer review of "Body Image among Girls in Indonesia: Associations with Disordered Eating Behaviors, Life Engagement, Desire for Cosmetic Surgery and Psychosocial Influences"

_ijerph, 2023, doi:10.3390/ijerph20146394_

Round 1

Reviewer 1 Report

The introduction does not provide  background about body esteem.

What was the criteria for separating those age groups?

Author Response

RESPONSE TO REVIEWER ONE

We thank Reviewer 1 for their review. Regarding the background of the study in relation to body esteem, we apologize that this was not made clear from the outset. The study examines body image among Indonesian adolescents and utilizes the Body Esteem Scale for Children and the Body Esteem Scale for Adults and Adolescents as a means to assess body image. We have modified the manuscript in several ways to make this distinction clear:

  • We have replaced ‘Body esteem’ in the title to ‘body image’.
  • In the Abstract, we describe what we mean by body esteem at first mention of this (i.e., a holistic measure of body image assessing the degree of satisfaction with one’s appearance, weight and shape) lines 12-13. This is again reiterated on lines 89-90 when the first mention of the body esteem scale is mentioned.
  • In the Method, we’ve made it clearer that the Body Esteem Scale is being used as a measure of body image (lines 134)
  • At the start of the Discussion, this point is reiterated (lines 289-290)

With regards to why we separated the age groups of girls in this study, we found this an interesting comment and one that we acknowledge we did not address clearly enough in the manuscript. First, the reviewer raising this issue made us mindful of the many instances in the manuscript with which we refer to ‘adolescent girls’ despite many of the girls in our study being preadolescent.  As such, we have removed ‘adolescent girls’ from the title of the manuscript (instead opting for ‘girls’) and amended the terminology ‘adolescent girls’ to ‘girls’ to more accurately capture who we are referring to throughout the manuscript.

Consideration of the two different age groups in this study is now fully explained in the Introduction to the study and reflected on in the Discussion. In summary, the little research to date examining body image among Indonesia has focused on girls of at least adolescent age (typically aged 14 years +). It is important to also consider if concerns are present in a younger cohort to help guide preventative efforts. Research beyond Indonesia suggests body image concerns present in girls younger than 14 and thus we wanted to examine that in the Indonesian context, alongside bolstering what is already known among an older cohort.

Lines 67-72: Furthermore, research to date [in Indonesia] exclusively focuses on older (adolescent) girls and young women. In other contexts, including other parts of Asia [16], research shows that girls can begin to experience body image concerns from early adolescence. Thus, an understanding of when body image concerns emerge among Indonesian girls is an important consideration, in order to identify the optimal time for preventative action.

Lines 121-127: Further, we examined these research aims separately among younger (10-13 years) and older girls (14-17 years). Research to date in Indonesia has exclusively focused on adolescent girls, however research from other contexts suggest body image concerns present earlier than this [4]. Understanding if such concerns, and their relationship with key health behaviours and psychosocial influences, are present among a younger cohort would be useful to guide if and when to intervene with preventative efforts in the Indonesian context.

Lines 463-466: The study also suggests preventative efforts to tackle body image concerns in Indonesia may be useful among preadolescents, with the findings indicating that concerns are already established in this age group.

Reviewer 2 Report

The relevance of the topic is unequivocal. The presented review of the research on the issue of body image is well structured and entails a large number of relevant papers.

The study was correctly conducted, the criteria were selected thoroughly, the quality of the results processing is high, which allowed the authors to draw well-reasoned conclusions.

It is worth noting that the authors expressed their gratitude to the respondents, which testifies to high standards of ethics and looks professional.

As a wish to the authors – a graphical representation of the model (the conception of the study) would certainly be a credit to the work.

Questions:

• Was the link between body image and the financial situation of the family identified?

• The questionnaire had a question about the height and weight of the girls. Were any results about the impact of the height and weight on self-esteem and other factors under study obtained?

Author Response

We thank Reviewer 2 for their positive appraisal of our work.

As we are testing associations and not a causal model, we are hesitant to add a graphical representation of the Tripartite Model to the manuscript.  On this point, we defer to the Editor.

Question 1 [Was the link between body image and the financial situation of the family identified?]: We decided against exploring the association between participants’ household income and body image due to the large number of factors this paper already examines. We chose to focus on psychosocial factors as these are potentially modifiable targets for intervention efforts, as opposed to social economic status which is much more difficult to change. Household income was measured to ensure we obtained a representative sample of girls from across Indonesia.

Question 2: With regards to considering the impact of height and weight (i.e., BMI) on self-esteem and other factors considered in the current study, this was considered beyond the scope of the current manuscript. The basis of the paper was to explore relationships specifically in relation to body image and thus considering additional relationships not focused on body image was deemed beyond the scope of the present manuscript.

Reviewer 3 Report

Dear authors,

your work might be interesting but the data was collected in 2017. many things could have changed in 6 years.

Author Response

RESPONSE TO REVIEWER THREE

Many thanks for your review. Thank you for raising the timing of data collection as a concern. We recognise the importance of emphasising this to contextualise our results and accordingly, have added a limitation in the Discussion section to highlight the caution to be taken due to the data being collected in 2017 (lines 411-415). However, we believe the associations we present are likely still as relevant now as they were in 2017 given these variables have been theorised since 1999 (Thompson et al, 1999) and remain the basis of much theorising about body image to this day (e.g., Roberts, Maheux, et al., 2022).

Discussion lines 453-454: Third, it should be noted that the data were collected in 2017 and thus might not be fully reflective of Indonesian girls’ body image and related factors today.

You have indicated that the introduction and cited references must be improved. Beyond what has been improved upon based on the feedback from the other two reviewers and the Editor, please can you elaborate what on what they think would strengthen this manuscript with regards to this?